# Influence of Achalasia on the Spirometry Flow–Volume Curve and Peak Expiratory Flow

**DOI:** 10.3390/diagnostics14090933

**Published:** 2024-04-29

**Authors:** Jelena Jankovic, Branislava Milenkovic, Aleksandar Simic, Ognjan Skrobic, Arschang Valipour, Nenad Ivanovic, Ivana Buha, Jelena Milin-Lazovic, Natasa Djurdjevic, Aleksandar Jandric, Nikola Colic, Stefan Stojkovic, Mihailo Stjepanovic

**Affiliations:** 1Clinic for Pulmonology, University Clinical Center of Serbia, 11000 Belgrade, Serbia; jjelena1984@gmail.com (J.J.); milenbra@gmail.com (B.M.); ivanabuha33@gmail.com (I.B.); natalidjurdjevic@yahoo.com (N.D.); jandric.alexander@gmail.com (A.J.); 2Medical Faculty, University of Belgrade, 11000 Belgrade, Serbia; apsimic65@gmail.com (A.S.); skrobico@gmail.com (O.S.); milinjelena@gmail.com (J.M.-L.); drcola12@gmail.com (N.C.); 3Clinic for Digestive Surgery, University Clinical Centre of Serbia, 11000 Belgrade, Serbia; nekic85@gmail.com; 4Karl Landsteiner Institute for Lung Research and Pulmonary Oncology, Klinik Floridsdorf, Vienna Health Care Group, 1210 Vienna, Austria; arschang.valipour@gesundheitsverbund.at; 5Institute for Medical Statistics and Informatics, University of Belgrade, 11000 Belgrade, Serbia; 6Center for Radiology and MR, University Clinical Center of Serbia, 11000 Belgrade, Serbia; 7Clinic for Gastroenterohepatology, University Clinical Center of Serbia, 11000 Belgrade, Serbia; stefanstojkovic@ymail.com

**Keywords:** achalasia, spirometry, PEF, flow–volume curve

## Abstract

Background: Achalasia is an esophageal motor disorder characterized by aperistalsis and the failure of the relaxation of the lower esophageal sphincter. We want to find out whether external compression or recurrent micro-aspiration of undigested food has a functional effect on the airway. Methods: The aim of this research was to analyze the influence of achalasia on the peak expiratory flow and flow–volume curve. All of the 110 patients performed spirometry. Results: The mean diameter of the esophagus was 5.4 ± 2.1 cm, and nine of the patients had mega-esophagus. Seven patients had a plateau in the inspiratory part of the flow–volume curve, which coincides with the patients who had mega-esophagus. The rest of the patients had a plateau in the expiration part of the curve. The existence of a plateau in the diameter of the esophagus of more than 5 cm was significant (*p* 0.003). Statistical significance between the existence of a plateau and a lowered PEF (PEF < 80) has been proven (*p* 0.001). Also, a statistical significance between the subtype and diameter of more than 4 cm has been proved. There was no significant improvement in the PEF values after operation. In total, 20.9% of patients had a spirometry abnormality finding. The frequency of the improvement in the spirometry values after surgery did not differ significantly by achalasia subtype. The improvement in FEV1 was statistically significant compared to the FVC values. Conclusions: Awareness of the influence of achalasia on the pulmonary parameters is important because low values of PEF with a plateau on the spirometry loop can lead to misdiagnosis. The recognition of various patterns of the spirometry loop may help in identifying airway obstruction caused by another non-pulmonary disease such as achalasia.

## 1. Introduction

Achalasia is a relatively rare gastrointestinal disorder without a clearly defined etiological factor of origin with low incidence. According to the literature, the incidence of achalasia is 1.6 in 100,000, and its prevalence is 10 per 100,000. It is more common in patients between 30 and 60 years [1,2]. Prevalence is nearly equal for both sexes, with a slight predominance in men, but this is most likely due to lifestyle (alcohol use, smoking, habits) [3]. There is a higher rate of achalasia in Europe and North America compared to Asia and the rest of the world [2]. Data from numerous studies show that incidence and prevalence have not changed in the last half century, although there are limitations due to the lack of larger study cohorts [1,2].

Achalasia is a neurodegenerative esophageal motor disorder characterized by a loss of peristalsis and the failure of the relaxation of the lower esophageal sphincter (LES). Because of the constriction of LES, food contents in the esophagus cannot be transported into the stomach, and gastrointestinal symptoms are more than four times more frequent. Numerous studies have examined the connection between the subtype of achalasia and the type of gastrointestinal symptoms. Dominant symptoms are regurgitation, dysphagia, and weight loss. The first two are the most prevalent symptoms in subtype 2 achalasia, and chest pain is dominant in subtype 3. Weight loss can lead to serious malnutrition [4]. The regurgitation and incomplete transport of undigested food, in future, can lead to aspirations, causing acute or chronical lung changes. The most common findings on chest computed tomography (CT) are fibrosis, ground glass opacification, pneumonia, atelectasis, or pulmonary abscess [5]. Aspiration also causes respiratory symptoms and structural or functional lung abnormalities [6]. Achalasia should be diagnosed as early as possible so that complications can be prevented and treated. In the end-stage of achalasia when mega-esophagus occurs, esophageal motility is irreversible, and complications are frequent [7]. Mega-esophagus develops in about 10% of inadequately treated patients; they have a worse prognosis, and in these patients, there is a possibility that the final therapeutic treatment will be esophagectomy [8].

There are two mechanisms of the effect of the dilated esophagus on the lungs in patients with achalasia. First is the external compression by the dilated esophagus, and the second is a recurrent micro-aspiration of undigested food causing bronchial mucosal inflammation [9]. Those mechanisms result in respiratory symptoms like stridor, dry cough, chest pain, or wheezing [10]. All of these can be similar to asthma or chronic obstructive pulmonary disease (COPD) symptoms. In addition, due to compression or bronchial mucosal inflammation, those patients can suffer the obstruction of spirometry, finding lower values of the parameters for small airways. This is because the wrong treatment with bronchodilators, without effect, can delay the correct diagnosis of achalasia for many years [11]. It is very important to not only measure spirometry parameters to verify the obstruction or restriction pulmonary findings but also to ensure that the flow–volume loop is analyzed first. Extrinsic tracheal compression of the dilated esophagus on the bronchial tree in patients with achalasia may cause a flattened expiratory flow–volume loop in the spirometry test. This plateau in the flow–volume curve is proof of intrathoracic disorder [12]. Several studies have showed that after operation, the asthmatic-like symptoms disappeared; the spirometry findings in pulmonary volumes improved, and in the flow–volume loop, improvements were found [13]. This demonstrates that achalasia should be considered to be one of the differential diagnoses of pulmonary disorders, especially the obstructive one.

According to the Chicago classification version 4 from 2021, when using high-resolution manometry, achalasia can be categorized into three subtypes [14]. Subtype 3 is very different from the other two; it is a spastic form of achalasia. Subtype 1 is an end-stage subtype with the complete loss of peristalsis, and subtype 2 occurs with intermittent periods of pan-esophageal pressurization [14]. The most common is subtype 2, and this has the best treatment outcomes [15]. Different types of achalasia have different prognoses, predominance of different gastrointestinal symptoms, ways of treatment, and outcomes.

Only a few studies describe the impact of achalasia on respiratory function, i.e., spirometry findings. However, there are insufficient data to correlate clinical manifestation, the diameter of esophagus, subtypes of achalasia, and the influence on the spirometry flow–volume loop.

In this article, we emphasize the influence of achalasia on peak expiratory flow (PEF) and the flow–volume curve, because numeric values of spirometry can lead us onto the wrong path of pulmonary diagnosis.

## 2. Materials and Methods

### 2.1. Study Group

A total of 110 patients who met the inclusion criteria were enrolled in this study. Patients eligible for this study were patients with an established diagnosis of achalasia surgically treated in the University Clinical Center of Serbia during the period from 2015 to 2020. The inclusion criteria for this study were patients with primary achalasia based on endoscopic, radiographic, and manometric findings and without specific previous treatment for achalasia. The exclusion criteria were all other gastrointestinal disorders such as gastroesophageal reflux disease or pseudo-achalasia because of esophageal carcinoma or patients who cannot undergo pulmonary tests.

In all subjects, age, sex, smoking status, body mass index, past medical history and comorbidity, smoking status, and gastrointestinal symptoms were recorded from medical history and interview. They were followed-up for 90 days after surgery.

The Ethics Committee in Belgrade approved this study. All subjects obtained and signed their informed consent before enrollment (following the principles of the International Declaration of Helsinki).

### 2.2. Radiological Assessment

Chest CT scan is the main radiological imaging method used because chest X radiography is not sensitive to the diagnosis of achalasia, and it can be presented only as a dilated shadow of mediastinum that can be caused due to many mediastinal disorders (lymphoma, carcinoma, retrosternal thyroid gland changes, aortic aneurism, or others).

Chest CT scan can show the presence of achalasia and exclude pseudo-achalasia because of carcinoma, and it can also describe esophageal morphology, esophageal dilatation, and the diameter of esophagus or mega-esophagus. It can also describe pulmonary structural findings [5,6]. All CT examinations were performed within 30 days before surgery (Siemens edge, PA, USA). The evaluation of the CT findings was led by a radiologist with more than 12 years of experience in this field working in the Center for Radiology and NMR, University Clinical Center of Serbia.

### 2.3. Endoscopy and Functional Diagnostics Assessment

Although manometry is the gold standard for the diagnosis of achalasia, esophagogastroduodenoscopy has an important role to play in early evaluation to exclude other diseases such as esophageal cancer, proximal gastric cancer, and eosinophilic esophagitis [16]. The endoscopic finding for achalasia, established by the Japan Esophageal Society, are as follows: dilatation of the esophageal lumen; retention of food and/or liquid; thickening of the esophageal mucosal; abnormal esophageal contraction waves; and functional stenosis of the esophago-gastric junction which fails to dilate by insufflation [16,17]. Based on esophageal manometry, which is the most sensitive test and gold standard, achalasia is categorized in three subtypes: subtype 1—with minimal esophageal pressurization and aperistalsis, subtype 2—with rapidly propagated compartmentalized pressurization across the entire length, and subtype 3—with spastic distal esophageal contractions [14].

### 2.4. Functional Spirometry Assessment

All patients from the study group performed spirometry before surgery as a preoperative preparation and postoperatively for up to 3 months from surgery to checkup. The follow-up period was 90 days for spirometry tests because we wanted to see whether the spirometry finding was definitive or just because of the influence of a dilated esophagus, to give time for the resolution of mucosal airway inflammation or the normalization of the airway wall and structure after the surgically reduced volume of the dilated esophagus. That timeframe of three months was optimal for the collection of definitive results.

Spirometry was in accordance with the guidelines of the European Respiratory Society (ERS) and performed on JAEGER^®^ MasterScreen Pneumo (USA) [18]. The forced expiratory volume in the first second (FEV1), forced vital capacity (FVC), FEV1/FVC ratio, and PEF were measured on the spirometer (Master Screen Body). Obstruction is defined by a post-bronchodilation FEV1/FVC ratio less than 70%. While restrictive disorder is defined if the FEV1/FVC ratio is greater than 70%, the FVC is lower than 80% of the predicted values [18].

All the pulmonary tests met the reproducibility and acceptability norms of the ERS guidelines, and the graph flow–volume curves did not show errors in the performance of the patients [18]. If there were errors or the patient did not adequately perform the tests for some reason, the findings were not interpreted or included in the overall statistical processing. Those patients were excluded from the study.

The typical finding of the plateau in the expiratory part of the flow–volume loop is presented in Figure 1. The flow–volume loop is a graphic display of airflow against lung volumes during the maximum inspiratory and expiratory maneuver. The plateau can be caused by the intra- or extra-thoracic influence on the airways. The first step in the interpretation of the spirometry findings is to describe the loop, and if this does not correlate with the ERS guidelines, we cannot interpret the volumes and capacity.

The Empey index was calculated as ratio of FEV_1_ (mL) and PEF (l/min) [19]. This index shows obstruction on the level of the upper respiratory tract or distal with great sensitivity according to the literature data. There is still no evidence of an influence or correlation with achalasia with this parameter in previous or recent studies.

### 2.5. Statistical Analysis

Analyses were performed in IBM SPSS Statistics for Windows, version 22.0 (Armonk, NY 2019, USA). For figure presentation, we used MedCalc for Windows, version 19.4 (Ostend, Belgium). The statistical significance was set at *p* < 0.05. In this study, descriptive and analytical statistical methods were used. The results were presented as mean values (MVs) with ± standard deviation (SD). Graphical and mathematical procedures were used to test the normality of distribution. Numerical data were compared with Student’s *T*-test for independent samples. Nominal data were compared with the Chi square test. The choice of the analytical statistical methods depended on the type of data and the distribution.

## 3. Results

The study was conducted as a retrospective study on 110 patients with confirmed achalasia treated in the Clinic for Digestive surgery, University Clinical Center of Serbia, for a period of 5 years. Achalasia was confirmed by manometry, and the subtype of achalasia was also determined in this way. Spirometry was performed on all patients in the Clinic for Pulmonology, University Clinical Center of Serbia.

In 41 (36.7%) patients, achalasia subtype 1 was diagnosed, in 56 (50.7%) patients, subtype 2 was diagnosed, and in only 13 (12.6%) patients was subtype 3 diagnosed using manometry.

The mean age of the patients was 52.8 years. No statistically significant difference was found in the average age between the different subtype groups of patients. Sex representation did not differ significantly according to the achalasia subtype, even though there was a higher prevalence of males in subtype 1 and females in subtype 3 (Table 1).

Smoking status also did not differ significantly according to the achalasia subtypes.

The mean diameter of the dilated esophagus was 5.4 ± 2.1 cm (range from 2.8 to 10.5 cm), and nine of them had mega-esophagus (more than 6 cm diameter). Esophageal diameter was widest in subtype 3.

The results of the one-way analysis of variance showed that the difference in the diameter of the esophagus was significant between subtype 1 and 3 (*p* = 0.011), as well as between subtype 2 and subtype 3 (*p* = 0.011).

In total, 23 patients, before surgery, had functional changes in spirometry, 13 of them with obstruction and 10 with restrictive finding. The other patients from the study group had normal spirometry tests. The respiratory function parameters were compared according to the subtype of achalasia; the results are shown in Table 2. There was no statistically significant difference in the pathological findings of respiratory function according to the subtype of achalasia (*p* 0.129).

The frequency of the improvement in respiratory function after surgery did not differ significantly by achalasia subtype (*p* 0.906).

The average values (percentage of predicted values) of the preoperative and postoperative spirometry parameters (FVC and FEV1) were compared. The improvement in FEV1 was statistically significant compared to the slight improvement in the average FVC values (*p* 0.007), but not in the FEV1/FVC ratio. The results are shown in Table 3.

Table 4 shows the frequency of the plateaus according to the subtype of achalasia. In seven patients, a plateau in the inspiration part on the spirometry flow–volume curve was described, which coincides with patients who had mega-esophagus, while the rest of the patients had a plateau in the expiration part of the curve. The existence of a plateau in the diameter of the esophagus of more than 5 cm was significant (*p* 0.003).

In addition, an Empey index of more than 10 was correlated with those seven (6.4%) patients with a plateau in the inspiratory part of the spirometry flow–volume curve. They all had a diameter of more than 6 cm; there was no significance for subtypes 1 or 2.

There was no influence of the presence of mega-esophagus or achalasia subtype on the appearance of the spirometry plateau. However, the statistical significance between the existence of a plateau and a lowered PEF (PEF < 80) has been proven (*p* 0.001) in Table 5.

There was no influence of the duration of symptoms on PEF, but the statistical significance between the subtype and a diameter of more than 4 cm is proved in Table 6.

There was no significant improvement in the PEF values after surgery.

## 4. Discussion

To our knowledge, this is the first study that evaluates the influence of a dilated esophagus on the spirometry curve and some of their values. The present study indicates that the existence of achalasia and dilated esophagus is a significant parameter for the existence of a plateau in the spirometry curve and decreased PEF values in a well-defined study population. These results strengthen the validation and expected values of these spirometry characteristics. This could be a screening tool for intrathoracic esophageal disorders with an impact on the airways.

The lack of data on the topic of improving pulmonary function tests and impact on the spirometry loop is due to the fact that only a few studies have been conducted on this topic. To our knowledge, this is the first study to test the impact of a dilated esophagus on the spirometry flow–volume loop, and only two studies exist with results about the impact on PEF values. This spirometry flow–volume loop is very useful for pulmonologists and provides information about the inspiratory limb (detect extra-thoracic upper airway obstruction) or expiratory limb (detect intra-thoracic airway obstruction) [20]. Based on abnormalities on the flow–volume curve, we refer patients for additional pulmonary function tests like impulse oscillometry, body plethysmography, or invasive examinations such bronchoscopy.

A study by Gupta and colleges, performed more than 10 years ago, showed a significant improvement in the pulmonary function parameters in their spirometry findings (FEV1, FVC, PEF) after LES dilation in patients with achalasia [21]. The limitation of this study is that it was conducted on only 38 patients. The previously mentioned study on the connection between the existence of achalasia and lung function disorders proved functional abnormalities in about 20% of their population group [9]. Our data are in accordance with those data with 20.9% pathological spirometry findings. The frequency of the improvement in respiratory function after surgery did not differ significantly between achalasia subtypes, but the improvement in FEV1 was statistically significant compared to the slight improvement in the average FVC values, and there was no improvement in the FEV1/FVC ratio. The explanation for this is the underlining mechanism of the occurrence of an obstructive or restrictive finding. Unlike obstructive lung diseases such as COPD or asthma, where there are structural reasons for air trapping and reduced flow through a bronchial tree, in achalasia, the mechanism is totally different [22,23]. In patients with achalasia, the reason is strictly mechanical due to the compression of the dilated esophagus, and for this reason, after surgery, the barrier that reduced the flow through the airway was physically removed. On the other hand, the repeated micro-aspiration of undigested food with gastric acid into the airways and lung parenchyma causes the worsening of chronic inflammation, which leads to surfactant damage, with consequent collapse and the development of fibrosis [23,24]. This reduces lung elasticity and volume at the expense of FVC values. Due to this permanent structural destruction of lung architecture and impaired diffusion, there is no significant improvement in FVC values after surgery. The FEV1/FVC ratio is important to explain the obstructive or restrictive findings [18]. Because there is not a classical broncho-dilatation effect as in obstructive diseases, and because less than 10% of all patients had obstructive values, this is an explanation why is not significant the improvement in the FEV1/FVC ratio. The FEV_1_ accounts for the greatest part of the exhaled volume and reflects the mechanical properties of the large- and medium-sized airways. FEV1 reduction is due to the increased airway resistance to expiratory flow [25]. Those are airways that had dilated esophagus influence, which did not occur on small airways. After the surgery and after being released from the mechanical compression on the flow in those airways, resistance was decreased, flow through the airways was improved, and the FEV1 values were higher. This is an explanation for the better results that were obtained in FEV1 after surgery and not in other spirometry values.

There was no difference in the improvement in lung function in relation to the subtype, although it was to be expected that it would improve more in the first two subtypes of achalasia. It is known that subtype 3 of achalasia has a hypertrophic wall of the esophagus that dilates less, unlike the first two subtypes, which are characterized by a thinned, irreversibly expanded wall and a greater degree of dilatation [26]. However, the explanation for the non-predominance of the first two subtypes is that in our group of respondents, the largest diameter was precisely in subtype 3. That is why there is no statistically significant difference.

In our study, there was no significant improvement in PEF values after the operation, but the statistical significance between the subtype and diameter of the esophagus of more than 4 cm with PEF was proved. Lower PEF was obtained in subtypes 1 and 2, perhaps because there were significantly more patients in these groups. In addition, patients with a mega-esophagus over six cm in diameter belonged to these two groups. Those patients do not have spirometry broncho-obstruction as a reason for lower PEF values (such as in COPD); instead, they have compression on the airways with dilated esophagus, and that is the reason for the lower PEF values. Patients in our study who had a flattened expiratory curve had a reduction in PEF (less than 80% predicted values). A study by Kossoyvaki and colleagues defined the possibility of monitoring patients with relapsed benign tracheal stenosis with PEF [27,28]. They showed great results. That could possibly be the case for patients with achalasia and a way of monitoring those patients. Maybe this parameter in future can be one of the markers for monitoring patients with achalasia or patients with mega-esophagus.

Esophageal compression can lead to changes in lung function parameters and the appearance of a plateau in the flow–volume curve [12,29]. A plateau on the spirometry flow–volume curve was verified in 38 of our patients. Patients with a plateau in the inspiration part coincided with patients who had mega-esophagus with subtype 1 and 2, while the rest had a plateau in the expiration part of curve. The appearance of a plateau in the expiratory part was expected considering that this is a characteristic of intrathoracic events with an impact of dilated esophagus on the airways both symptomatically and functionally [19,20]. Given that it is almost impossible for the esophagus to compress the cartilaginous part of the airways and lead to narrowing, and because of its anatomical position, it could compress the posterior trachea wall. In this way, we could explain the similarity of the spirometry findings in patients with achalasia due to the compression of the dilated esophagus on the posterior membranous wall with patients with tracheomalacia or EDAC (excessive dynamic airway collapse). EDAC is a rare disorder and is characterized by excessive flexion of the posterior tracheal wall membrane into the airway lumen during exhalation [30]. In this way, this can consequently lead to respiratory symptoms such as dyspnea, cough, and frequent respiratory infection or pneumonia. Also, asthma or COPD unresponsiveness to inhaled therapy may often be misdiagnosis, or dyspnea symptoms may increase after therapy [31]. For this reason, in addition to EDAC and tracheomalacia, with these characteristics of potentially pulmonary disorder, we should also think of achalasia when we find dose symptoms and a flattened expiratory curve. Bronchoscopy is needed for those patients in order to give a conclusion, and upper gastrointestinal endoscopy is needed to exclude or confirm one or another disorder.

However, the main question is the reason for the appearance of a plateau in the inspiratory part of the spirometry loop in six patients. That is unexpected for the spirometry curve in achalasia. The inspiratory plateau is dominant for extra-thoracic disorders in upper airway disorders or the influence on them (changes in the vocal cords, enlarged thyroid glands). An otolaryngologist examined all six patients, and it was concluded that the findings were normal. The explanation for this could be the existence of a mega-esophagus, which dilates over time, and the proximal part of the esophagus, which acquires a sigmoid shape in the upper aperture. This is a characteristic of the irreversible end-stage form of achalasia with a diameter of more than 6 cm [32]. In this way, we could explain the inspiratory plateau resulting from the compression of the massively dilated proximal part of the esophagus at the level of the upper thoracic aperture. That could be an explanation also for the Empey index values over 10 in these patients [11,33]. There is a high sensitivity and specificity of this index, with values greater than 10 for the detection of obstruction at the level of the upper respiratory tract [33]. Many studies have been written on this topic, but none of them describe the influence of the proximal esophagus on the respiratory thoracic aperture, as well as for the Empey index values over 10 in these patients [11,33]. This index rises as the obstruction becomes more severe [34]. This is one more reason to keep in mind achalasia in differential diagnoses.

The fact that we evaluated and strengthened the internal validity of this study could help us to think differentially about the diagnosis of achalasia if patients have a plateau in the expiratory part of the spirometry loop, low PEF, and a combination of gastrointestinal and respiratory symptoms. Spirometry data often do not correlate with the severity of respiratory problems.

This study had some limitations. Firstly, there is a low number of patients with subtype 3, though this matches the data from the world literature. The second limitation is that only one surgical center is included. The third limitation is that a certain percentage of patients were not motivated to undergo the pulmonary function tests control after surgical treatment, especially patients with normal values of respiratory function findings. Also, another limitation is that the number of patients within certain groups (subtype 3, patients with PEF less than 80%, improvement in PEF) is small, so the interpretation of such results is difficult, and no correlation with other parameters or prediction is possible. The MRC scale is a good example that demonstrates the subjective feeling of dyspnea and is everyday practice for COPD patients, but it was not performed in this study, so perhaps it could be considered for those patients in the future.

## 5. Conclusions

The flow–volume loop is a simple, easily available, and noninvasive test of lung function. The recognition of various patterns of the spirometry loop may help in identifying airway obstruction caused by another non-pulmonary disease such as achalasia. A persistently flattened abnormal flow–volume loop appearance and elevated Empey’s index should act to prompt the consideration of an airway cause for breathing difficulties in patients presenting with exertional breathlessness but without obstructive pulmonary diseases.

## Figures and Tables

**Figure 1 diagnostics-14-00933-f001:**
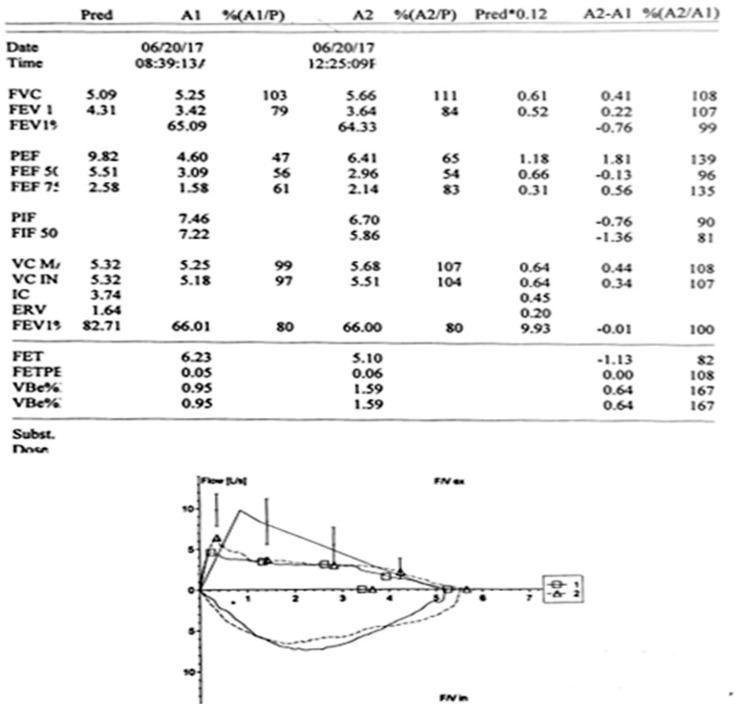
Plateau on expiratory part of flow–volume loop.

**Table 1 diagnostics-14-00933-t001:** Comparison of basic characteristics of patients according to type of achalasia.

Subtype		1	2	3	*p*
Age (years)		51 ± 17.2	51.6 ± 17.4	54.9 ± 18	0.757
Sex *n* (%)	Male	24 (58.5)	28 (50)	5 (38.5)	0.200
	Female	17 (41.5)	28 (50)	8 (61.5)	
Smoking status *n* (%)	Non-smokers	23 (56)	30 (54.5)	6 (42.8)	0.918
	Ex-smokers	7 (17.1)	9 (16.1)	2 (14.1)	
	Smokers	11 (26.9)	17 (29.4)	5 (38.1)	

**Table 2 diagnostics-14-00933-t002:** Respiratory function findings and correlation of the presence of improvement after surgery.

Subtype		1 *n* = 41	2 *n* = 56	3 *n* = 13	*p*
Spirometry findings *n* (%)	normal	31 (75.6)	46 (82.2)	10 (76.9)	0.129
	obstruction	3 (7.3)	7 (12.5)	3 (23.1)	
	restriction	7 (17.1)	3 (5.3)	0 (0)	
Improvement after surgery *n* (%)	no	28 (68.2)	37 (66.1)	9 (69.2)	0.906
	yes	13 (31.8)	19 (33.9)	3 (30.8)	

**Table 3 diagnostics-14-00933-t003:** Comparison of the difference in average preoperative and postoperative spirometry parameters.

	Preoperative Value	Postoperative Value	*p*
FVC (% of predicted)	67.1 ± 2.1	68.8 ± 1.7	0.05
FEV1 (% of predicted)	66.7 ± 4.3	78.3 ± 3.8	0.007
FEV1/FVC (% of predicted)	80.5 ± 6.6	81.2 ± 4.7	0.453

**Table 4 diagnostics-14-00933-t004:** The frequency of the existence of a plateau according to the subtype of achalasia.

Subtype	1	2	3
*n* (%)	17(40.5)	18(31.5)	3(21.4)
Expiratory curve *n* (%)	12 (70.6)	16 (88.9)	3 (100)
Inspiratory curve *n* (%)	5 (29.4)	2 (11.1)	0

**Table 5 diagnostics-14-00933-t005:** Correlation between the existence of a plateau and PEF.

		PEF < 80			
		0.00		1.00	
		Count	Column *N* %	Count	Column *N* %
Plateau *n*	no	5	30.0%	70	75.5%
	yes	13	70.0%	22	24.5%

**Table 6 diagnostics-14-00933-t006:** Correlation between subtype, diameter, and duration of symptoms and PEF.

PEF < 80		
	*N* (%)	*p*
Subtype		
1	7 (6.4)	
2	10 (9.1)	0.006
3	4 (3.6)	
Diameter (>4 cm)	17 (15.5)	0.021
Duration of symptoms (years)		0.786

## Data Availability

The data that support the findings of this study are available from the corresponding author (MS) upon reasonable request.

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
