# Peer review of "Influence of Achalasia on the Spirometry Flow–Volume Curve and Peak Expiratory Flow"

_diagnostics, 2024, doi:10.3390/diagnostics14090933_

Round 1

Reviewer 1 Report

Comments and Suggestions for Authors

Minor considerations

- The aim of the Abstract and the end of the Introduction differ slightly; please harmonize them.

- Insert number of subjects in the Abstract.

- What is the purpose of Figure 1? It seems unnecessary.

Line 247: where it says "DES" shouldn't it say "LES"?

Major considerations

Were pulse oximetry and six-minute walk test data collected on the study participants?

Were any of the six patients who had an inspiratory plateau smokers or ex-smokers? Couldn't patients with altered pre-surgical spirometry have hindered the analysis?

Reviewer 2 Report

Comments and Suggestions for Authors

Thank you for the possibility to review the manuscript titled: “Influence of achalasia on the spirometry flow-volume curve and PEF”. The study involves 110 patients with achalasia that underwent respiratory evaluation before and after surgery. The design, material and methods section is well organized and easy to understand. The literature includes most of the available studies on the topic. Overall, this is an interesting study, however, there are several minor recommendations:

-“Background\Aims” in the abstract. Please delete “aims” as you mention aim in the material and methods section. There is no need for repetition.

-“Gender” is a psychological term. Please change it to “sex”, which is a biological term.

-Figure 1 is of poor quality, please consider changing it

-Please check the manuscript for possible type errors and review the language of the manuscript

-Please consider adding in the discussion section that spirometry data often does not correlate with the severity of respiratory problems. MRC scale is a good example that demonstrates subjective feeling of dyspnea. Otherwise include this point in the limitation section.

Please take into consideration the recommendations in the spirit of improving the quality of the submission.

Comments on the Quality of English Language

Requires minor editing
